# Deep Active Speech Cancellation with Mamba-Masking Network

## Abstract

We present a novel deep learning network for Active Speech Cancellation (ASC), advancing beyond Active Noise Cancellation (ANC) methods by effectively canceling both noise and speech signals. The proposed Mamba-Masking architecture introduces a masking mechanism that directly interacts with the encoded reference signal, enabling adaptive and precisely aligned anti-signal generation—even under rapidly changing, high-frequency conditions, as commonly found in speech. Complementing this, a multi-band segmentation strategy further improves phase alignment across frequency bands. Additionally, we introduce an optimization-driven loss function that provides near-optimal supervisory signals for anti-signal generation. Experimental results demonstrate substantial performance gains, achieving up to 7.2dB improvement in ANC scenarios and 6.2dB in ASC, significantly outperforming existing methods.

## 1 Introduction

Active Noise Cancellation (ANC) is a critical audio processing technique aimed at eliminating unwanted noise by generating an anti-noise signal (Lueg, 1936; Nelson & Elliott, 1991; Fuller et al., 1996; Hansen et al., 1997; Kuo & Morgan, 1999). ANC has practical applications in improving hearing devices for individuals with hearing impairments and reducing chronic noise exposure, thereby mitigating hearing loss risks. It also enhances focus, productivity, and listening experiences while reducing stress. Traditional ANC algorithms, like LMS and its deep learning variants (Zhang & Wang, 2021; Park et al., 2023; Mostafavi & Cha, 2023; Cha et al., 2023; Pike & Cheer, 2023; Singh et al., 2024), have been widely adopted. However, these methods face limitations when dealing with more complex and high-frequency audio signals, as they are primarily designed to target noise. This paper addresses Active Speech Cancellation (ASC), which expands upon ANC by targeting the cancellation of both noise and speech signals. To our knowledge, this is the first work to actively cancel both noise and speech using deep learning, setting it apart from existing methods and enabling new research directions.

We propose a novel Mamba-Masking multi-band architecture that applies a masking mechanism to the encoded signal. This facilitates precise anti-signal generation, enhancing phase alignment and improving ANC performance. This design is particularly effective for speech signals, as it accounts for their broader frequency spectrum. Coupled with an optimization-driven loss function, this approach achieves improved performance in dynamic acoustic scenarios. Results demonstrate up to a 7.2 dB improvement in ANC and a 6.2 dB gain in ASC for speech signals, outperforming deep-learning based baselines, which are considered state-of-the-art in the field.

## 2 Related Work

### 2.1 Active Noise Cancellation

The concept of ANC was first introduced by Lueg Lueg (1936), focusing on sound oscillation cancellation. Given that ANC algorithms must adapt to variations in amplitude, phase, and noise source movement (Nelson & Elliott, 1991; Fuller et al., 1996; Hansen et al., 1997; Kuo & Morgan, 1999), most ANC algorithms are based on the Least Mean Squares (LMS) algorithm (Burgess, 1981), which is effective in echo cancellation. The FxLMS algorithm extends LMS by using an adaptive

filter to correct distortions in primary and secondary paths. Boucher et al. (1991) examined errors in FxLMS due to inaccuracies in estimating the secondary path inverse, where nonlinearities affect performance. Solutions such as the Filtered-S LMS (FSLMS) (Das & Panda, 2004), which uses a Functional Link Artificial Neural Network (FLANN) (Patra et al., 1999), and the Volterra Filtered-x LMS (VFXLMS) (Tan & Jiang, 2001), which employs a multichannel structure, address these issues. The Bilinear FxLMS (Kuo & Wu, 2005) improves nonlinearity modeling, and the Leaky FxLMS (Tobias & Seara, 2005) introduces a leakage term to mitigate overfitting. The Tangential Hyperbolic Function-based FxLMS (THF-FxLMS) (Ghasemi et al., 2016) models saturation effects for enhanced performance. Gannot & Yeredor (2003) proposed blind source separation for noise cancellation. Moreover, Oppenheim et al. (1994) proposed single channel ANC based on Kalman filter formulation (Revach et al., 2021) and Rafaely (2009) investigated spherical loudspeaker arrays for local sound control.

ANC using deep learning was first proposed by Zhang & Wang (2021) with a convolutional-LSTM network for estimating both amplitude and phase of the canceling signal $y(n)$. Recurrent CNNs were later explored by Park et al. (2023); Mostafavi & Cha (2023); Cha et al. (2023), autoencoder-based networks Singh et al. (2024), and fully connected neural networks, were also applied to the problem Pike & Cheer (2023). Shi et al. (2020; 2022b); Park & Park (2023); Shi et al. (2023a); Luo et al. (2022; 2023b;a; 2024b) have developed methods that select fixed-filter ANC (SFANC) from pre-trained control filters to achieve fast response times. Concurrently, Zhu et al. (2021); Shi et al. (2022a); Zhang & Wang (2023); Shi et al. (2023b); Antoñanzas et al. (2023); Xiao et al. (2023); Zhang et al. (2023b); Shi et al. (2024) advanced multichannel ANC systems. Luo et al. (2023c) introduced a CNN-based approach for real-time ANC, further enhanced with Kalman filtering. Zhang et al. (2023a) incorporated an attention mechanism for real-time ANC using the Attentive Recurrent Network (ARN)(Pandey & Wang, 2022). Other significant real-time ANC contributions include genetic and bee colony algorithm-based methods (Ren & Zhang, 2022; Zhou et al., 2023).

## 2.2 ACTIVE SPEECH CANCELLATION

ASC has been explored in various studies, each employing different approaches to predict and cancel unwanted speech signals. Kondo & Nakagawa (2007) introduced an ASC method using a Linear Predictive Coding (LPC) model to predict the speech signal for generating the canceling signal $y(n)$. Donley et al. (2017) took a different approach by controlling the sound field to cancel speech using a linear dipole array of loudspeakers and a single microphone, effectively reducing the speech signal in the target area. Iotov et al. (2022) employed a long-term linear prediction filter to anticipate incoming speech, enabling the cancellation of the speech signal. Additionally, Iotov et al. (2023) proposed HOSpLP-ANC, which combines a high-order sparse linear predictor with the LMS algorithm for effective speech cancellation.

## 2.3 MAMBA ARCHITECTURE

Recently, the Mamba architecture has been introduced (Gu & Dao, 2023; Dao & Gu, 2024), leveraging State Space Models (SSMs) to achieve notable improvements in various audio-related tasks. One of the key advantages of the Mamba architecture is its ability to perform fast inference on sequences up to a million in length, marking a significant improvement over traditional generative models. This has enabled advancements in several applications, including automatic speech recognition (Zhang et al., 2024b;a), speech separation (Jiang et al., 2024a; Li & Chen, 2024), speech enhancement (Chao et al., 2024; Luo et al., 2024a; Quan & Li, 2024), speech super-resolution (Lee & Kim, 2024), sound generation (Jiang et al., 2024b), audio representation (Shams et al., 2024; Yadav & Tan, 2024; Erol et al., 2024), sound localization (Xiao & Das, 2024; Mu et al., 2024), audio tagging (Lin & Hu, 2024), and deepfake audio detection (Chen et al., 2024).

## 3 BACKGROUND

The feedforward ANC system consists of reference and error microphones, a loudspeaker, and two acoustic transfer paths: the primary path $P(z)$, from the noise source to the error microphone, and the secondary path $S(z)$, from the loudspeaker to the error microphone. The signal captured by the reference microphone is denoted as $x(n)$, while the signal captured by the error microphone is

| Method | $[\mathbf{y}^*, \mathbf{y}]$ | $[\mathbf{P} * \mathbf{x}, \mathbf{S} * \mathbf{y}]$ | $[\mathbf{S} * \mathbf{y}^*, \mathbf{S} * \mathbf{y}]$ | $[\mathbf{P} * \mathbf{x}, \mathbf{S} * \mathbf{y}^*]$ |
|---|---|---|---|---|
| - NOAS | -9.85 | -16.53 | -18.56 | -23.63 |
| + NOAS | -12.77 | -17.60 | -19.62 | |

Table 1: Comparison of NMSE distances for different objectives, with and without NOAS optimization. Measured on DeepASC training set.

Figure 1: Typical feedforward ANC system diagram.

denoted as $e(n)$. These signals are processed by the ANC controller to produce a canceling signal $y(n)$, which is emitted by the loudspeaker $f_{LS}$. The loudspeaker output $f_{LS}\{y(n)\}$, after passing through the secondary path $S(z)$, generates the anti-signal denoted by $a(n)$. The relationship is described by: $a(n) = S(z) * f_{LS}\{y(n)\}$. Similarly, the reference signal $x(n)$, transmitted through the primary path $P(z)$, generates the primary $d(n)$, which is expressed as: $d(n) = P(z) * x(n)$.

The error signal $e(n)$ is the difference between the primary signal $d(n)$ and the anti-signal $a(n)$:

$$e(n) = d(n) - a(n) \tag{1}$$

The goal of the ANC controller is to minimize the error signal $e(n)$, ideally to zero, indicating successful noise cancellation. In the feedback ANC approach, only the error signal $e(n)$ is utilized to generate the canceling signal, aiming to minimize residual noise at the error microphone.

One of the widely used metrics for measuring noise attenuation in ANC is the Normalized Mean Square Error (NMSE) between two signals, defined by:

$$\text{NMSE}\left[\mathbf{u}, \mathbf{v}\right] = 10 \cdot \log_{10}\left( \frac{\sum_{n=1}^{M}(u(n) - v(n))^2}{\sum_{n=1}^{M} u(n)^2} \right) \tag{2}$$

where $\mathbf{u}$ and $\mathbf{v}$ are the vector representations of the signals $u(n)$ and $v(n)$ such that $\mathbf{u} = [u(1), ..., u(M)]$ and $\mathbf{v} = [v(1), ..., v(M)]$. Here, $M$ represents the total number of samples. Typically, $u(n)$ refers to the target signal, while $v(n)$ denotes the estimated signal. A lower NMSE value indicates a better estimation, reflecting a closer alignment between the estimated signal and the target signal. In the context of ANC, typically $u(n)$ is the primary signal $d(n)$, while $v(n)$ will be the anti-signal $a(n)$. A schematic representation of the ANC system is illustrated in Figure 1.

## 4 METHOD

We propose a novel architecture that integrates the Mamba framework (Gu & Dao, 2023) with a multi-band masking strategy based on Dual-path Mamba blocks (Jiang et al., 2024a). A filter bank splits the input, and each band is processed by an encoder-masker-decoder pipeline. An improved cancellation accuracy is achieved by using a new loss function, which uses a near-optimal anti-signal as ground truth. A diagram of the proposed architecture is shown in Fig. 2.

### 4.1 DEEPASC ARCHITECTURE

Let $x(n)$ be the reference signal such that $1 \leq n \leq M$. The reference signal $x(n)$ is decomposed into $Q \in \mathbb{N}$ different frequency bands $x_1(n), \ldots, x_Q(n)$. These frequency bands are evenly divided such that for the maximum frequency $F$, the $i$-th frequency band $x_i(n)$ covers the frequency range $\left[(i-1)\frac{F}{Q}, i\frac{F}{Q}\right]$ where $1 \leq i \leq Q$. In addition to the decomposed bands, the original full-band signal $x(n)$ is included as $x_0(n)$. Each band $x_i(n)$ (where $0 \leq i \leq Q$, the zero index is for the entire unfiltered band) is then processed through its own Masking-Band block (MB-block). Each MB-block comprises an encoder and a masking network that utilize Mamba-based layers. Within each MB-block, the encoder consists of a one-dimensional convolution layer $E_i$ with a kernel size $k$ and a stride of $k/2$. The encoder transforms the $i$-th reference signal $x_i(n)$ into a two-dimensional latent representation:

$$\mathbf{H}_i = E_i[\mathbf{x}_i] \tag{3}$$

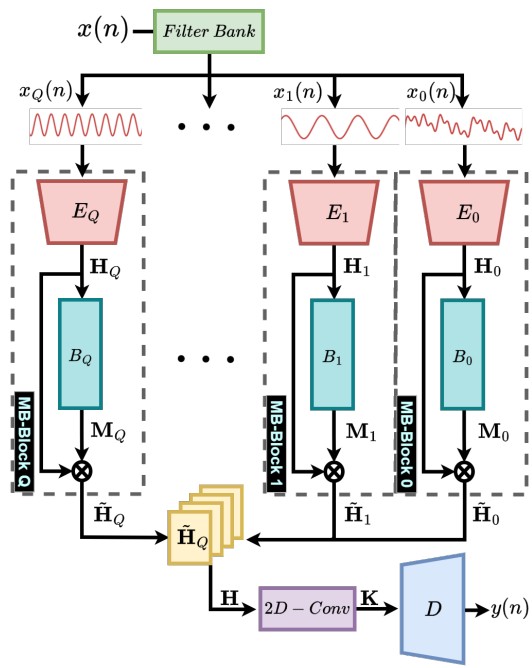

Figure 2: DeepASC Architecture.

Table 2: NMSE focusing on VAD-masked speech-active segments ($\eta^2 = 0.5$).

| Method | TIMIT ($\downarrow$) | WSJ ($\downarrow$) | LibriSpeech ($\downarrow$) |
|---|---|---|---|
| DeepANC | -9.7 | -7.93 | -12.63 |
| ARN | -8.2 | -5.61 | -12.29 |
| **DeepASC** | **-17.8** | **-15.56** | **-17.66** |

Table 3: Average NMSE ($\downarrow$) for ANC methods on noise and speech, evaluated on real-world measured **P** & **S** with $\eta^2 = 0.5$.

| Method | Factory ($\downarrow$) | Babble ($\downarrow$) | WSJ ($\downarrow$) |
|---|---|---|---|
| DeepANC | -9.29 | -10.94 | -8.26 |
| ARN | -8.97 | -11.17 | -10.70 |
| **DeepASC** | **-12.09** | **-13.87** | **-12.23** |

Table 4: FLOPs & NMSE comparison for different ANC methods.

| Method | FLOPs (G) ($\downarrow$) | NMSE ($\downarrow$) |
|---|---|---|
| DeepANC | 7.199 | -10.69 |
| ARN | 5.281 | -11.61 |
| **DeepASC** | **2.419** | **-13.46** |

where $\mathbf{H}_i \in \mathbb{R}^{B \times C}$, with $B = \frac{M-k}{\frac{k}{2}} + 1$, $C$ representing the number of channels after the convolution operator and $\mathbf{x}_i$ is the vector representation of $x_i(n)$. The latent representation $\mathbf{H}_i$ is then passed through the Mamba-based layers $B_i$ to produce the $i$-th masking signal $\mathbf{M}_i$:

$$\mathbf{M}_i = B_i[\mathbf{H}_i] \tag{4}$$

The MB-blocks estimates $Q+1$ masks of the same latent dimension $\mathbf{M}_i \in \mathbb{R}^{B \times C}$. These masks are element-wise multiplied with the encoder outputs $\mathbf{H}_i$ to produce masked hidden representations $\tilde{\mathbf{H}}_i$:

$$\tilde{\mathbf{H}}_i = \mathbf{H}_i \cdot \mathbf{M}_i \tag{5}$$

Then, the masked hidden representations $\tilde{\mathbf{H}}_i$ is concatenated over all frequency bands $i$, such that:

$$\mathbf{H} = concat\left[\tilde{\mathbf{H}}_0, ..., \tilde{\mathbf{H}}_Q\right] \tag{6}$$

Where $\mathbf{H} \in \mathbb{R}^{(Q+1) \times B \times C}$. The hidden tensor $\mathbf{H}$ is then processed with a 2D convolution layer with a kernel size of $1 \times 1$ and one output channel that produces $\mathbf{K} \in \mathbb{R}^{B \times C}$. To obtain the vector representation of the canceling signal $\mathbf{y}$, we apply a decoder $D$. Specifically, the decoder is a one-dimensional transpose convolutional layer with a kernel size $k$ and a stride of $k/2$. This decoder ensures that the canceling signal $\mathbf{y}$ has the same dimensions as the reference signal $x(n)$:

$$\mathbf{y} = D[\mathbf{K}], \tag{7}$$

where $\mathbf{y} = [y(1), \ldots, y(M)]$ is the vector representation of the canceling signal $y(n)$, and $M$ is the length of the signal.

### 4.2 OPTIMIZATION OBJECTIVE

The training protocol for the proposed method consists of two distinct phases: (i) ANC loss minimization, and (ii) near optimal anti-signal fine-tuning optimization. Each phase employs the NMSE loss function (Eq. 2) but with different optimization objectives.

**ANC Loss:** In the first phase, the optimization aims to minimize the residual error signal. Given a reference signal $x(n)$ and the model output $y(n)$, the error loss function is defined as follows:

$$\mathcal{L}_{\text{ANC}} = \text{NMSE}\left[\mathbf{P} * \mathbf{x}, \mathbf{S} * f_{LS}\{\mathbf{y}\}\right] \tag{8}$$

where $\mathbf{P}$ and $\mathbf{S}$ represent the vectorized forms of the primary-path impulse response $P(z)$ and the secondary-path impulse response $S(z)$, respectively; $\mathbf{x}$ and $\mathbf{y}$ are the vectorized forms of the reference signal $x(n)$ and the canceling signal $y(n)$. The operator $*$ denotes convolution. Both $\mathbf{P}$ and $\mathbf{S}$ are obtained from the simulator employed in our study.

**Near Optimal Anti-Signal Optimization (NOAS):** A key challenge in formulating ANC as a supervised learning task is designing a training objective function that accounts for the influence of the acoustic paths (Zhang & Wang, 2021). In practice, the model output $y(n)$ is nonlinearly transformed by $f_{LS}$ and filtered through $S(z)$, with the goal of minimizing the residual error $e(n)$. However, when $S(z)$ attenuates certain frequency bands that are not attenuated by $P(z)$, conventional loss functions (e.g., Eq. 8) penalize the model despite producing optimal pre-propagation anti-noise. This mismatch introduces misleading gradients, destabilizing training and hindering convergence.

To address this challenge, we propose the NOAS loss function. The NOAS loss symmetrically incorporates the secondary path $S(z)$ on both sides of the NMSE calculation. Specifically, each reference signal $x(n)$ is associated with its NOAS target $y^*(n)$. To determine the near-optimal anti-signal $y^*(n)$, we employ a gradient descent-based algorithm during a pre-processing stage. This stage operates over each example, solving the following optimization problem for each reference signal $x(n)$ separately:

$$\mathbf{y}^* = \arg\min_{\tilde{\mathbf{y}}} \text{NMSE}\left[\mathbf{P} * \mathbf{x}, \mathbf{S} * f_{LS}\{\tilde{\mathbf{y}}\}\right] \tag{9}$$

where $\mathbf{y}^*$ is the near-optimal anti-signal. The optimization starts with a random anti-signal and iteratively adjusts it to minimize the NMSE for the given reference signal $x(n)$. The resulting near-optimal anti-signal $y^*(n)$ is then used to form the target during the fine-tuning stage. In particular, the near-optimal anti-signal $y^*(n)$ is used to define the following loss function:

$$\mathcal{L}_{\text{NOAS}} = \text{NMSE}\left[\mathbf{S} * f_{LS}\{\mathbf{y}^*\}, \mathbf{S} * f_{LS}\{\mathbf{y}\}\right] \tag{10}$$

Table 1 reports empirical measurements from the DeepASC training set that support our approach. Following the first training phase, the NMSE between $[\mathbf{P} * \mathbf{x}, \mathbf{S} * \mathbf{y}]$ is 7.1 dB higher than that between $[\mathbf{P} * \mathbf{x}, \mathbf{S} * \mathbf{y}^*]$, indicating the model retains significant capacity for further optimization. Additionally, the NMSE between $[\mathbf{S} * \mathbf{y}^*, \mathbf{S} * \mathbf{y}]$ is 2.03 dB lower than that between $[\mathbf{P} * \mathbf{x}, \mathbf{S} * \mathbf{y}]$, suggesting that learning the NOAS target $y^*$ is more tractable than direct cancellation from $x$. Note that the optimization occurs in the $\mathbf{S}$-projected space, rather than directly in the canceling signal space (i.e. NMSE $[\mathbf{y}^*, \mathbf{y}]$). For a comprehensive explanation of this design choice and related measurement observations, see Appendix A.

## 5 EXPERIMENTS

### 5.1 DATASETS

The training data is sources from the AudioSet dataset (Gemmeke et al., 2017), which we encompassed 248 diverse audio classes including hubbub, speech noise, and babble. A total of 22,224 audio samples (approximately 18.5 hours) were standardized to 3 seconds and resampled to 16kHz, following the settings of the ARN method (Zhang et al., 2023a). Of these, 20,000 samples (90%) were used for training and 2,224 for testing. Additional test data were sourced from the NoiseX-92 dataset (Varga & Steeneken, 1993), which includes noise types such as bubble, factory, and engine noise. To evaluate speech generalization, we incorporated test samples from three speech corpora: TIMIT (Garofolo, 1993) (24 speakers across 8 dialects), LibriSpeech (Panayotov et al., 2015) (40 audiobook speakers), and WSJ (Garofolo et al., 1993) (8 speakers reading news text).

### 5.2 SIMULATOR

Following prior studies (Zhang & Wang, 2021; Zhang et al., 2023a), we simulate a rectangular enclosure with dimensions $[3, 4, 2]$ meters (width, length, height). Room impulse responses (RIRs) are generated using the image-source method (Allen & Berkley, 1979) via the Python `rir_generator` package (Habets, 2006), with a high-pass filter enabled and RIR length fixed at 512 taps. Microphone and speaker positions are as follows: error microphone at $[1.5, 3, 1]$ m, reference microphone at $[1.5, 1, 1]$ m, and cancellation load speaker at $[1.5, 2.5, 1]$ m. During training,

reverberation times are randomly sampled from $\{0.15, 0.175, 0.2, 0.225, 0.25\}$ seconds; testing uses a fixed reverbration time of $0.2$ seconds. To model loudspeaker saturation, we adopt the Scaled Error Function (SEF) (Tobias & Seara, 2006), commonly used in ANC research (Zhang & Wang, 2021; Zhang et al., 2023a; Mostafavi & Cha, 2023; Cha et al., 2023), defined as: $f_{SEF}\{y\} = \int_0^y e^{-\frac{z^2}{2\eta^2}} dz$ where $y$ denotes the loudspeaker input and $\eta^2$ controls nonlinearity intensity. The SEF approximates linearity as $\eta^2 \to \infty$, and behaves like a hard limiter as $\eta^2 \to 0$, effectively simulating saturation constrained by physical loudspeaker limits.

### 5.3 HYPERPARAMETERS

An extensive grid search and cross-validation were employed to determine the optimal hyperparameters for each method. The hyperparameter values reported here correspond to the configurations that achieved the best performance in our experimental setup. The DeepASC architecture was trained with $Q = 2$, where the full-band employed a medium (M) configuration with 16 layers, and each of the two sub-bands used a small (S) configuration with 8 layers. The bands decomposition filters are generated using the $scipy.signal.firwin$ function and applied to the signal via $torch.conv1d$. The temporal duration $M$ was set to 48,000 samples, corresponding to 3-second audio signals sampled at 16 kHz. The channel dimension $C$ was set to 256, and the kernel size $W$ was defined as 16. A batch size of 2 was used for training the DeepASC architecture. The Adam optimizer (Diederik, 2014) was employed with an initial learning rate of $1.5 \times 10^{-4}$. A learning rate decay factor of 0.5 was applied every 2 epochs after an initial warm-up period of 30 epochs. Gradient clipping with a threshold of 5 was applied to prevent exploding gradients.

### 5.4 BASELINE METHODS

We compared our proposed method against several established ANC techniques, including Deep-ANC (Zhang & Wang, 2021), Attentive Recurrent Network (ARN) (Zhang et al., 2023a), Filtered-x LMS (FxLMS), and Tangent Hyperbolic Function FxLMS (THF-FxLMS) (Ghasemi et al., 2016). All methods were evaluated under identical simulation settings in both linear and nonlinear scenarios, using noise and speech signals. FxLMS, DeepANC, and ARN were implemented and trained by us, employing the same dataset used for our model. DeepANC used 20-ms STFT frames with 10-ms overlap; ARN used 16-ms frames with 8-ms overlap. Our implementations reproduced results consistent with the original papers. These baselines were chosen to ensure a comprehensive comparison across both classical adaptive filtering and recent deep-learning-based ANC paradigms.

## 6 RESULTS

### 6.1 NOISE CANCELLATION

Table 5 presents the NMSE results for ANC algorithms under engine, factory, and babble noise using 3-second segments from NoiseX-92. Each model was evaluated with and without nonlinear distortions ($\eta^2 = \infty$, 0.5, and 0.1). For traditional methods (FxLMS and THF-FxLMS), gradient clipping at $1e^{-4}$ and step sizes of 0.05 (engine), 0.4 (factory), and 0.3 (babble) were used to ensure stability. The results indicate that these methods performed worse than deep learning-based approaches.

Among deep-learning-based models, and without considering the nonlinearity saturation effect, DeepASC achieved state-of-the-art performance. Without nonlinearity ($\eta^2 = \infty$), DeepASC outperformed ARN by 4.29dB, 4.64dB, and 7.26dB for engine, factory, and babble noise, respectivly. With moderate distortion ($\eta^2 = 0.5$), DeepASC yielded respective improvements of 4.36dB, 4.62dB, and 7.13dB. Under severe distortion ($\eta^2 = 0.1$), DeepASC led with a margin of 3.79dB for engine noise, 4.4dB for factory noise, and 5.76dB for babble noise. Figures3a, 3b, and 3c illustrate that DeepASC consistently outperforms ARN, DeepANC, and FxLMS across all timesteps.

The proposed method was also evaluated for speech enhancement in the presence of noise using active noise cancellation. The PESQ and STOI metrics, presented in Table 6, compare the performance of DeepANC, ARN, and DeepASC (w/o NOAS) across various SNR levels in the presence of factory noise with nonlinear distortion of $\eta^2 = \infty$. The results demonstrate that DeepASC outperforms ARN, showing improvements in PESQ scores by $0.7$, $0.92$, and $0.84$ at SNR levels of 5dB, 15dB, and 20dB, respectively. A similar trend is observed for STOI, with enhancements of $0.08$, $0.03$, and $0.02$ for the same SNR levels.

Table 5: Average NMSE (↓) in dB for DeepASC and other algorithms across various noise types and nonlinear distortions. Lower values indicate better performance.

| Method/Noise type | Engine (↓) | | | Factory (↓) | | | Babble (↓) | | |
|---|---|---|---|---|---|---|---|---|---|
| $\eta^2$ | $\infty$ | 0.5 | 0.1 | $\infty$ | 0.5 | 0.1 | $\infty$ | 0.5 | 0.1 |
| FxLMS | -3.38 | -3.33 | -3.32 | -3.27 | -3.17 | -3.11 | -5.39 | -5.33 | -5.30 |
| THF-FxLMS | - | -3.37 | -3.36 | - | -3.26 | -3.24 | - | -5.39 | -5.36 |
| DeepANC | -13.96 | -13.91 | -13.6 | -10.7 | -10.69 | -10.62 | -12.42 | -12.4 | -12.22 |
| ARN | -14.59 | -14.59 | -14.38 | -11.61 | -11.61 | -11.54 | -12.91 | -12.9 | -12.72 |
| **DeepASC** | -18.88 | -18.95 | -18.17 | -16.25 | -16.23 | -15.94 | -20.17 | -20.03 | -18.48 |

Table 6: Average NMSE (dB), STOI and PESQ for deep ANC models in noisy speech situations with LS nonlinearity ($\eta = 0.5$) and factory noise at different SNR levels.

| Method | Noise only | SNR = 5dB | | SNR = 15dB | | SNR = 20dB | |
|---|---|---|---|---|---|---|---|
| | NMSE (↓) | STOI (↑) | PESQ (↑) | STOI (↑) | PESQ (↑) | STOI (↑) | PESQ (↑) |
| DeepANC | -10.69 | 0.83 | 1.39 | 0.93 | 2.10 | 0.96 | 2.45 |
| ARN | -11.61 | 0.84 | 1.51 | 0.94 | 2.43 | 0.96 | 2.92 |
| **DeepASC** | -15.94 | 0.92 | 2.21 | 0.97 | 3.35 | 0.98 | 3.76 |

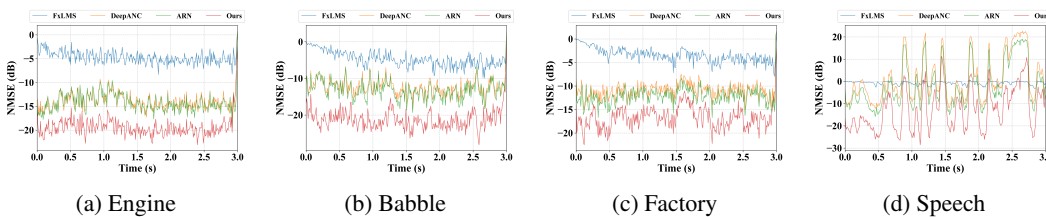

|  (a) Engine | (b) Babble | (c) Factory | (d) Speech |

Figure 3: Comparison of NMSE (dB) over time for different noise types.

## 6.2 SPEECH CANCELLATION

Table 7 presents the average NMSE values for different ANC algorithms across three speech datasets: TIMIT, LibriSpeech, and WSJ, with speech segments affected by varying levels of nonlinear distortions. As observed in the noise cancellation case, in speech cancellation, the non-deep learning methods—FxLMS and THF-FxLMS—demonstrate suboptimal performance compared to deep learning-based approaches. Among the deep learning methods, DeepASC achieves the best overall results, surpassing the other algorithms significantly.

In the case without nonlinear distortions ($\eta^2 = \infty$), DeepASC shows improvements over ARN by 6.13 dB, 4.78 dB, and 5.95 dB for the TIMIT, LibriSpeech, and WSJ datasets, respectively. In the presence of moderate nonlinear distortions ($\eta^2 = 0.5$), DeepASC continues to outperform ARN, with improvements of 6.18 dB for TIMIT, 4.34 dB for LibriSpeech, and 5.99 dB for WSJ. Under more severe nonlinear distortions ($\eta^2 = 0.1$), DeepASC maintains its superior performance, with enhancements of $5.97$dB, $2.46$dB, and $5.81$dB for TIMIT, LibriSpeech, and WSJ datasets, respectively. Figure 4 compares the power spectra and spectrograms of different ANC methods applied to a speech signal. DeepASC achieves significantly better noise suppression across all frequencies, particularly in the high-frequency range - DeepASC outperforming DeepANC and ARN. As shown in Figure 3d, DeepASC consistently yields lower NMSE across nearly all time steps,

However, it is worth noting that the standard NMSE metric may not fully reflect DeepASC's effectiveness in speech-active scenarios, due to its sensitivity to silent intervals within speech recordings. Examination of the spectrograms in Figure 4 and perceptual listening tests of the error signals highlight a qualitative performance gap between DeepASC and baseline methods that is not entirely captured by NMSE alone. To address this, we performed an evaluation using VAD-based masking to isolate speech-active regions, computing NMSE[VAD($\mathbf{P} * \mathbf{x}$), VAD($\mathbf{S} * \{f_{LS}\{\mathbf{y}\}\}$)] (see Appendix D for VAD mask examples and details). The results in Table 2 for the $\eta^2 = 0.5$ case reveal a more pro-

Table 7: Average NMSE ($\downarrow$) in dB for DeepASC and other algorithms across various speech datasets and nonlinear distortions. Lower values indicate better performance.

| Method/Dataset | TIMIT ($\downarrow$) | | | LibriSpeech ($\downarrow$) | | | WSJ ($\downarrow$) | | |
|---|---|---|---|---|---|---|---|---|---|
| $\eta^2$ | $\infty$ | 0.5 | 0.1 | $\infty$ | 0.5 | 0.1 | $\infty$ | 0.5 | 0.1 |
| FxLMS | -1.39 | -1.36 | -1.26 | -3.43 | -3.40 | -3.28 | -1.92 | -1.90 | -1.85 |
| THF-FxLMS | - | -1.37 | -1.35 | - | -3.41 | -3.39 | - | -1.91 | -1.89 |
| DeepANC | -8.52 | -8.56 | -8.48 | -11.92 | -11.81 | -11.08 | -7.54 | -7.55 | -7.51 |
| ARN | -10.31 | -10.27 | -10.2 | -12.87 | -12.74 | -11.87 | -9.48 | -9.48 | -9.42 |
| **DeepASC** | -16.44 | -16.45 | -16.17 | -17.65 | -17.08 | -14.33 | -15.43 | -15.47 | -15.23 |

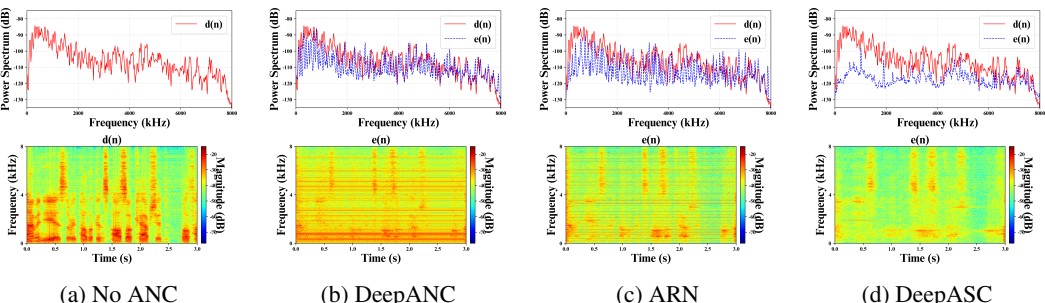

(a) No ANC          (b) DeepANC          (c) ARN          (d) DeepASC

Figure 4: Spectrograms and Power Spectra of Speech Signal (00da010c from WSJ) using Different ANC methods without nonlinear distortions ($\eta^2 = \infty$)

Table 8: Average NMSE ($\downarrow$) in dB for DeepASC and other algorithms under real-time constraints.

| Method | Runtime (S) | Engine ($\downarrow$) | Babble ($\downarrow$) | TIMIT ($\downarrow$) | LibriSpeech ($\downarrow$) | WSJ ($\downarrow$) |
|---|---|---|---|---|---|---|
| ARN | 0.0116 | -10.73 | -9.37 | -12.40 | -6.32 | -6.61 |
| DeepANC | 0.0125 | -11.13 | -9.84 | -12.18 | -7.61 | -8.33 |
| **DeepASC** | 0.0136 | -16.56 | -13.14 | -14.81 | -12.52 | -13.40 |

nounced advantage: DeepASC outperforms the alternative methods by 8.1,dB on TIMIT, 7.63,dB on WSJ, and 5.03,dB on LibriSpeech.

### 6.3 REAL-WORLD SIMULATION

We expanded our investigation to assess the performance of our method in real-world settings, testing it across various simulation scenarios. This was necessary because the fixed task acoustic setup, which relies on the image method, has limitations regarding generalizability and real-world performance. We utilized the dataset from Liebich et al. (2019), which includes acoustic paths from 23 individuals, measured in the real world and encompassing both primary and secondary paths. We applied DeepASC, along with baseline approaches, to the updated simulation conditions, evaluating their performance using Factory and Babble noise from the NoiseX-92 dataset and speech samples from the WSJ dataset. The results in Table 3 present the average NMSE across these categories. The results demonstrate that DeepASC consistently outperforms the alternative methods, achieving improvements of 2.80dB in Factory noise, 2.70dB in Babble noise, and 1.53dB on WSJ.

### 6.4 RUNTIME ANALYSIS

Real-time performance is critical in ANC systems. To ensure compliance with the causality constraint, we adopt a future-frame prediction strategy as employed in Zhang & Wang (2021); Zhang et al. (2023a). Let $T_p$ and $T_s$ denote the acoustic delays of the primary and secondary paths, respectively, and $T_{\text{ANC}}$ be the algorithm's processing time. The system must satisfy the constraint $T_{\text{ANC}} < T_p - T_s$, which evaluates in our setting to $T_{\text{ANC}} < \frac{2}{343} - \frac{0.5}{343} = 0.0043$s.

To meet this, DeepASC is optimized for edge deployment using a single S-band, NOAS optimization, and future prediction. Experimental results summarized in Table 8 show that while DeepANC

Table 9: Performance comparison across datasets for different DeepASC variants.

| Method | Factory ($\downarrow$) | TIMIT ($\downarrow$) | LibriSpeech ($\downarrow$) | WSJ ($\downarrow$) |
|---|---|---|---|---|
| DeepASC (31.9M) | -15.94 | -16.36 | -16.95 | -15.32 |
| DeepASC-1L Band (32M) | -15.72 | -15.95 | -16.64 | -15.32 |
| DeepASC-LSTM (33.4M) | -11.53 | -13.33 | -13.92 | -13.28 |
| DeepASC-Transformer (38M) | -14.93 | -14.01 | -15.2 | -13.76 |
| DeepASC-No-Dual (31.9M) | -11.59 | -12.36 | -13.07 | -12.3 |
| DeepASC-No-Masking (31.9M) | -3.36 | 7.43 | -6.82 | 1.37 |

and ARN yield slightly lower clock-time latencies (up to 2.6 ms), the difference is negligible due to all models relying on future frame prediction within a 0.01s window (160 samples at 16 kHz). It is demonstrated that DeepASC meets real-time constraints while achieving superior ANC performance related to other methods by NMSE margins of 5.43, 3.3, 2.41, 5.07, and 4.91 dB on engine noise, babble noise, TIMIT, WSJ, and LibriSpeech, respectively.

The computational complexity of the models was additionally assessed by comparing their FLOPs, averaged across 20 three-second samples from the Noisex-92 dataset, as presented in Table 4. The single-band, small variant of DeepASC demonstrated exceptional efficiency, requiring only 2.862G FLOPs while consistently surpassing the performance of the other models. This highlights its superior balance between computational cost and effectiveness.

## 6.5 ABLATION STUDY

We conducted ablation experiments to evaluate the impact of key components in DeepASC, including the Masking mechanism, Mamba layer, multi-band processing and dual-path structure. For the Mamba layer, we replaced it with either a Transformer or LSTM in the masknet. The Transformer-based model used 12 layers for the full-band and 6 layers for small bands, with 2 blocks per layer (d_model=256, 4 heads, d_ffn=1024). The LSTM variant employed the same depth and block structure, each block comprising two LSTM layers (hidden_dim=256). The original DeepASC uses Mamba blocks (d_model=256, ssm_dim=16, mamba_conv=4) with 16 layers for the full-band and 8 for small bands. For the masking mechanism, we eliminated it entirely, allowing direct anti-signal prediction. To test multi-band processing, we trained a single-band model with an equivalent parameter count to the 3-band model. To isolate the dual-path structure, we removed it while preserving activation shapes. Unless noted otherwise, all models used the 3-band configuration (one full-band M and two sub-band S paths), without NOAS optimization.

Results summarized in Table 9 (with $\gamma^2 = 0.5$) show the masking mechanism is crucial—its removal degrades performance by at least 10.13 dB (LibriSpeech). The Mamba block significantly outperforms alternatives: while the Transformer performs comparably, it lags by at least 1.01 dB (Factory); the LSTM model performs worse across all datasets. Finally, although the single-band model matches the 3-band setup on WSJ, the 3-band variant consistently outperforms it elsewhere, confirming the effectiveness of multi-band processing beyond parameter scaling. These findings support our architectural decisions and demonstrate the efficacy of DeepASC. Appendix B provides a further ablation study, focusing on the importance of NOAS optimization.

## 7 CONCLUSION AND LIMITATIONS

This paper introduced a novel ASC method based on the Mamba-Masking architecture. By decomposing and transforming the encoded signal through the masking, our model enhances anti-signal generation and phase alignment, leading to more effective cancellation. Combined with an optimization-based loss (NOAS), the approach achieves near-optimal performance, improving ANC and ASC by 7.2 dB and 6.2 dB respectively over state-of-the-art baselines on voice signals. These results underscore the Mamba-Masking Network's capacity to manage diverse frequencies and real-world acoustic conditions, where conventional models often under-perform. Despite empirical gains from components like Mamba layers and NOAS, a rigorous theoretical justification for their effectiveness remains an open question. Additionally, we have yet to fully exploit the Mamba architecture's long-context modeling capabilities. Overall, our framework addresses key limitations in current ANC systems, and opens new directions for advanced audio cancellation technologies.

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

## APPENDIX

## A    NOAS DESIGN CHOICES & MOTIVATION

As previously discussed, the optimization process is conducted in the $\mathbf{S}$-projected space rather than directly within the domain of the canceling signal itself—that is, by minimizing NMSE $[\mathbf{S} * \mathbf{y}^*, \mathbf{S} * \mathbf{y}]$ instead of NMSE $[\mathbf{y}^*, \mathbf{y}]$. This projection via $\mathbf{S}$ utilizes prior knowledge captured during the initial training phase, specifically the temporal dependencies embedded in the structure of $\mathbf{y}$. For the sake of clarity, we will use the mean squared error (MSE) as the distance metric. To illustrate this, consider the example depicted in Figure 5. In this illustrative case, we assume both $\mathbf{P}$ and $\mathbf{S}$ are defined as simple averaging filters (e.g., $[0.5, 0.5]$ for a two-dimensional signal). Let $\mathbf{y}^*$ denote the optimal anti-noise signal such that $\mathbf{P} * \mathbf{x} = \mathbf{S} * \mathbf{y}^* = 0$. Additionally, for the model's output signal $\mathbf{y}$, we have $\mathbf{P} * \mathbf{x} = \mathbf{S} * \mathbf{y} = 0$, which indicates that $\mathbf{y}$ is already optimal in the projected space. However, if we were to directly optimize in the native domain of $\mathbf{y}$ without regard to the projection, the resulting estimate $\mathbf{y}'$—although potentially closer to $\mathbf{y}^*$—might lead to suboptimal performance since $\mathbf{P} * \mathbf{x} \neq \mathbf{S} * \mathbf{y}'$.

This behavior is attributed to the properties of convolution with a fixed filter (in this case, $\mathbf{S}$), which does not constitute an isometry and thus fails to preserve distances in the original space. As such, optimization in the $\mathbf{S}$-projected space more faithfully reflects the desired performance criterion.

This conceptual rationale, together with the previously stated motivation, is further substantiated by the provided measurements in Table 1. In particular, the NMSE values were consistently lowest between $[\mathbf{S} * \mathbf{y}^*, \mathbf{S} * \mathbf{y}]$, lending strong support to the claim that $\mathbf{S} * \mathbf{y}^*$ constitutes a feasible optimization target for $\mathbf{S} * \mathbf{y}$. Additionally, a noteworthy observation arises following the NOAS optimization: the NMSE between $[\mathbf{P} * \mathbf{x}, \mathbf{S} * \mathbf{y}]$ is significantly reduced by 1.07 dB. This empirical finding challenges the notion that NOAS merely functions as a regularization term, instead indicating that it plays a more active role in enhancing the quality of the learned representations.

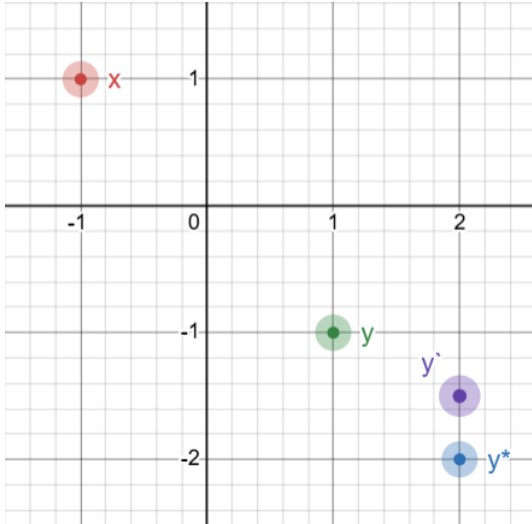

Figure 5: $\mathbf{S}$-projection importance visualization for NOAS optimization.

# B  ABLATION STUDY - 2ND PART

To assess the contributions of the NOAS fine-tuning optimization in our method, we conducted an ablation study focusing on multiband processing, band size (small vs. medium), and the impact of NOAS optimization on them. Table 10 presents the results of this analysis, reporting the NMSE across four datasets: Factory, TIMIT, LibriSpeech, and WSJ, all evaluated under nonlinear distortion conditions ($\eta^2 = 0.5$).

Table 10: Average NMSE ($\downarrow$) in dB for noise and speech using multiple variants of DeepASC, with nonlinear distortion of $\eta = 0.5$.

| Method/Dataset | Factory ($\downarrow$) | TIMIT ($\downarrow$) | LibriSpeech ($\downarrow$) | WSJ ($\downarrow$) |
|---|---|---|---|---|
| + S - MultiBand - NOAS | -13.46 | -14.26 | -14.88 | -13.20 |
| + S - MultiBand + NOAS | -14.19 | -14.54 | -15.24 | -13.55 |
| + M - MultiBand - NOAS | -15.19 | -15.82 | -16.56 | -14.86 |
| + M - MultiBand + NOAS | -16.09 | -16.25 | -16.92 | -15.27 |
| + MultiBand - NOAS | -15.94 | -16.36 | -16.95 | -15.32 |
| **Full Method** | -16.23 | -16.45 | -17.08 | -15.47 |

In our notation, "+ S - Multiband - NOAS" refers to a small band configuration (8 mamba layers) without multiband processing or NOAS optimization, while "+ S - Multiband + NOAS" refers to the same small band architecture with NOAS optimization applied. Similarly, "+ M - Multiband - NOAS" represents a medium band configuration (16 mamba layers) without NOAS, and "+ M - Multiband + NOAS" applies NOAS optimization to the same medium band model. The "+ Multi-Band - NOAS" is defined as a configuration that employs one full medium band and two small sub-bands without NOAS optimization applied, whereas the **Full Method** is defined as the same configuration with NOAS optimization applied.

All models were initially trained using the ANC loss function defined in Eq. 8. Configurations with "+ NOAS" were fine-tuned using NOAS optimization, whereas configurations with "- NOAS" were trained exclusively using the ANC loss in Eq. 8. The results demonstrate that the removal of NOAS optimization consistently degrades performance across all datasets. For instance, on the Factory dataset, applying NOAS optimization to the small band model leads to a performance improvement of 0.73dB, while the medium band model shows a larger improvement of 0.90dB. This trend holds across the other datasets, reinforcing the crucial role of NOAS optimization in enhancing model performance. Multiband processing further improves the overall effectiveness of DeepASC. The **Full Method** consistently outperforms the "+ Multiband - NOAS" configuration, with gains of 0.29dB, 0.09dB, 0.13dB, and 0.15dB on the Factory, TIMIT, LibriSpeech, and WSJ datasets, respectively. Interestingly, the performance of the "+ M - Multiband + NOAS" configuration is higher than that of the "+ Multiband - NOAS" variant by 0.15dB. This indicates that while multiband processing is valuable, the choice of band size plays a significant role in the model's performance, with larger band sizes, particularly when combined with NOAS, yielding the best results.

# C  MODEL ANALYSIS

The number of frequency bands in the DeepASC architecture is a critical hyperparameter affecting performance. Table 11 compares DeepASC's performance across different band configurations for the Factory noise, TIMIT, LibriSpeech, and WSJ datasets, with $\eta^2 = 0.5$. The "1-band" models use a single full band, while the "3-band" and "4-band" models incorporate one medium band with two and three smaller sub-bands, respectively. A 2-band model, which would require two full bands, was excluded as it falls outside the intended design of DeepASC.

As shown in Table 11, increasing the number of bands improves model performance. For example, the 4-band configuration outperforms the 3-band variation by 0.58 dB, 0.19 dB, 0.37 dB, and 0.48 dB on the Factory noise, TIMIT, LibriSpeech, and WSJ datasets, respectively. This enhancement comes from the model's improved focus on sub-frequency bands, benefiting higher frequencies.

Table 11: Average NMSE ($\downarrow$) in dB of our method (**w/o NOAS**) for Noise and Speech using different number of bands, with nonlinear distortion of $\eta^2 = 0.5$.

| Method/Dataset | #Bands | Factory ($\downarrow$) | TIMIT ($\downarrow$) | LibriSpeech ($\downarrow$) | WSJ ($\downarrow$) |
|---|---|---|---|---|---|
| DeepASC (small) | 1 | -13.46 | -14.26 | -14.88 | -13.22 |
| DeepASC (medium) | 1 | -15.19 | -15.82 | -16.56 | -14.86 |
| DeepASC | 3 | -15.94 | -16.36 | -16.95 | -15.32 |
| DeepASC | 4 | -16.52 | -16.55 | -17.41 | -15.84 |

**Model Size Comparison.** Table 12 compares model size and performance, with NMSE evaluated on factory noise under nonlinear distortion of $\eta = 0.5$. DeepASC variants in this comparison are without NOAS optimization. The results indicate that even the smallest DeepASC configuration (1-band, small) outperforms the ARN architecture by $1.85$ dB, despite using only half the parameters (8.0M vs. 15.9M). This is a significant outcome given the critical importance of model size in real-time ANC applications where latency is critical.

Table 12: Comparison of different deep learning based ANC methods based on parameter size.

| Models | #Params | NMSE ($\downarrow$) |
|---|---|---|
| Deep-ANC | 8.8M | -10.69 |
| ARN | 15.9M | -11.61 |
| DeepASC, 1 Band, S | 8.0M | -13.46 |
| DeepASC, 1 Band, M | 15.8M | -15.19 |
| DeepASC, 3 Bands | 31.9M | -15.94 |
| DeepASC, 4 Bands | 40.0M | -16.52 |

## D  VAD MASKS VISUALIZATION

We provide additional visualizations and a detailed explanation of the VAD mask employed in our proposed method. Specifically, the audio signals were segmented using a window length of 256 samples with an overlap of 128 samples. The energy threshold for the VAD was set to 10% of the maximum energy observed within the corresponding speech signal. Frames with energy values below this threshold were marked as inactive (i.e., masked). Figure 6 presents the VAD masks applied to 9 distinct speech samples. In each subplot, the red line indicates the binary VAD mask. Segments where the mask is zero correspond to suppressed (i.e., nulled) portions of the signal, whereas segments with a non-zero mask retain the original signal content unaltered.

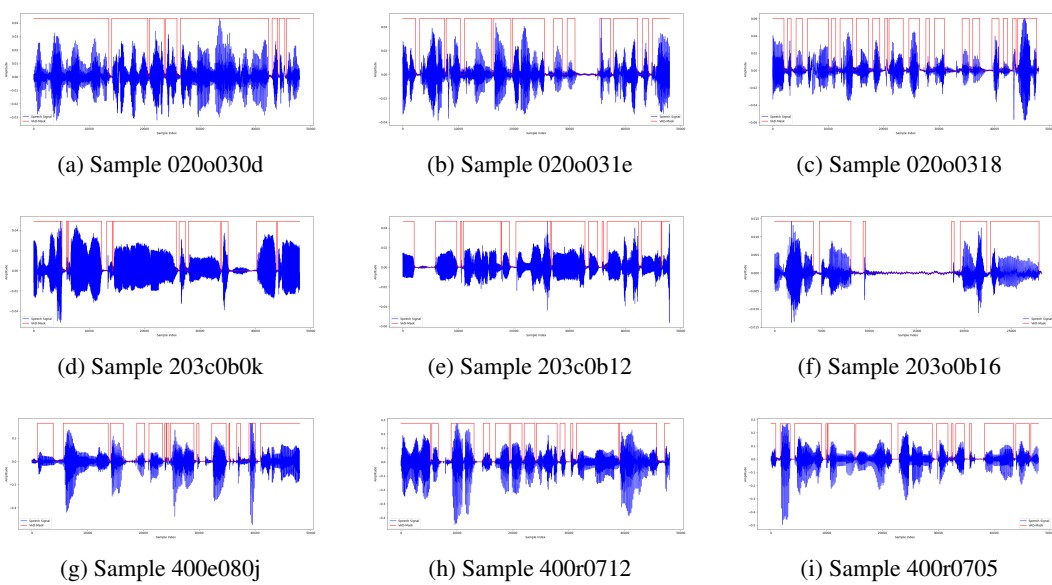

(a) Sample 020o030d  (b) Sample 020o031e  (c) Sample 020o0318

(d) Sample 203c0b0k  (e) Sample 203c0b12  (f) Sample 203o0b16

(g) Sample 400e080j  (h) Sample 400r0712  (i) Sample 400r0705

Figure 6: Visualization of VAD masks applied to nine different speech signals from WSJ dataset. Each subplot shows the energy contour with the overlaid red VAD mask.

