# OpenReview forum: "Deep Active Speech Cancellation with Mamba-Masking Network"
_ICLR.cc/2026/Conference — ICLR 2026 Conference Withdrawn Submission_

### Official Review · Reviewer_LJd5 · 2025-10-29

**Soundness:** 3
**Presentation:** 3
**Contribution:** 2
**Rating:** 2
**Confidence:** 3

**Summary:**

This paper proposes a deep learning framework for Active Speech Cancellation, that eliminates both environmental noise and
unwanted speech signals.The framework integrates three core components to enable accurate anti-signal generation:
- Frequency Band Decomposition: The input reference signal is decomposed into different bands.
- Mamba-Based Masker: Mamba layers generate a masking signal.
- Efficiency and Generalization: Balances high performance with low computational cost.

**Strengths:**

The experimental design is sound and adequately validates the main methodology. The application of the state space model (Mamba) in the context of active audio or noise cancellation appears to be a novel contribution.

The presentation is good, where the structure is well-organized, data are clearly presented through tables and figures, and technical details are sufficiently elaborated.

**Weaknesses:**

The work lacks significant originality:
- The Mamba-Masking mechanism adapts existing Mamba models rather than introducing a novel design.
- Multi-band processing is a commonly used technique in audio-related tasks.
- And the NOAS loss function merely refines existing formulations without addressing previously unsolved problems. The
proposed optimization strategy first derives an intermediate target y' based on the reference signal x, and then obtains the final
output y based on y'. Essentially, the approach remains grounded in reference signal-based loss minimization.
It primarily recombines established techniques rather than proposing a breakthrough framework for speech enhancement or
noise cancellation. Thus, the rationale behind the performance improvements and the overall significance of this approach
remain unclear and require further clarification

**Questions:**

- Only short audio samples were considered. In your studies, only short audios (3 second long) were tested. Have you tested the
     model on audio clips of different lengths? I'd like to know its actual performance on varying audio durations as well as how it
     compares to other models.

- Multi-band processing is a commonly used method in audio, and relevant references should be added to the article.

---

### Official Review · Reviewer_1LNE · 2025-10-31

**Soundness:** 3
**Presentation:** 2
**Contribution:** 3
**Rating:** 4
**Confidence:** 3

**Summary:**

Summary: This paper tackles the active speech cancellation (ASC) task, and proposes a mamba-masking architecture based on the subband design. And an optimization-driven loss is proposed to better cater for anti-signal generation.  Experiments on ANC and ASC scenarios demonstrate the sperioirty of the proposed method.

**Strengths:**

1. Acoustic scenario:  While existing literature concentrate on the active noise cancellation, this paper also considers the speech cancellation scenario, and such a case is seldomly tackled in previous DNN-based works.
2. Architecture: This work is the first to adopt Mamba as the modeling block, and a subband-based design is proposed, which achieves better performance than previous works.
3. This paper analyzes the limitation of existing ANC loss, and proposes a near optimal anti-signal loss, which the target signal convolved with the primary path is the replaced by its near optimal counterpart.

**Weaknesses:**

1. While this paper generalizes the original ANC scenario into the ASC, I think it belongs to the novely in acoustic setting, rather than in machine learning problem solving. Thus, I think it may be more suitable for acoustic/speech journals, e.g., JASA or TASLP.
2. While ANC has direct applications, I am not quite clear about the concrete applications of ASC, and I am a little curious whether this scenario setting is practical.
3. The motivaiton of using Mamba structure seems unclear.
4. While ANC/ASC is quite strict to processing delay and edge-implementation, I am suspicious of the proposd method since Mamba structure does not support CPU inference yet.

**Questions:**

1. It would be beneficial to clearly state the motivation for adopting deep learning to address ANC problems in the introduction section. For example, its capability to handle nonlinear distortions. This would make the introduction more compelling and better grounded.
2. It would be helpful to highlight that deep learning–based ANC methods are fixed-parameter systems, in contrast to traditional adaptive filtering approaches. This distinction would help clarify the methodological differences.
3. Latency is a critical issue for ANC methods, particularly for those implemented in the frequency domain. Besides model size and FLOPs, it is recommended to include an analysis of the algorithmic latency of the proposed method  on edge devices to provide a more comprehensive evaluation.

---

### Official Review · Reviewer_htfH · 2025-11-02

**Soundness:** 2
**Presentation:** 3
**Contribution:** 2
**Rating:** 2
**Confidence:** 3

**Summary:**

This paper introduces a novel deep learning framework for Active Speech Cancellation (ASC), which extends beyond traditional Active Noise Cancellation (ANC) by canceling both noise and speech. The proposed Mamba-Masking architecture employs a masking mechanism on the encoded reference signal to generate a precisely aligned anti-signal, even for rapidly changing, high-frequency speech. Combined with a multi-band segmentation strategy and an optimization-driven loss function, our method achieves significant performance gains, with up to 7.2 dB improvement in ANC and 6.2 dB in ASC, substantially outperforming existing state-of-the-art methods.

**Strengths:**

1) Integrates state-space Mamba layers with a multi-band masking architecture in an encoder–masker–decoder pipeline to target phase alignment and high-frequency dynamics in speech, which is a coherent architectural direction for active cancellation tasks.

2) Proposes a two-stage training with a “near-optimal anti-signal” target that projects supervision through the secondary path, a conceptually consistent way to mitigate misleading gradients from acoustic-path mismatch.

3) Presents broad empirical coverage with ablations, suggesting the masking mechanism and the Mamba blocks are key contributors, and reports consistent NMSE gains over selected deep ANC baselines across multiple noise and speech sets.

**Weaknesses:**

1) Novelty and positioning are unclear: the work is framed as the first to actively cancel both noise and speech with deep learning, but the paper neither cleanly delineates ASC from stronger ANC/beamforming/speech enhancement/wavefield control, nor demonstrates clear conceptual advances beyond “a stronger ANC system that also targets speech.”
2) Real-time causality inconsistency. the theoretical latency bound is stringent, yet reported runtimes exceed that bound and rely on future-frame prediction; the paper does not reconcile total pipeline latency, prediction lookahead, and acoustic path variations in a way that demonstrates robust real-time feasibility.
3) Metric design and bias: NMSE dominates, and the use of VAD-masked NMSE to focus on speech-active regions introduces a bespoke post-hoc evaluation that can bias comparisons; there is no statistical testing, limited perceptual evidence, and no task-level metrics aligned with ASC’s purported goals.
4) Notation and presentation rigor: symbol usage and definitions (e.g., kernels/strides, nonlinearity parameters, and projected-space objectives) need tighter, fully specified formulations and implementation details to ensure reproducibility and theoretical clarity.
5) This topic has been rare at ICLR, the deep-learning baselines are limited and feel dated (2016, 2021, 2023), while traditional ASC methods (e.g., LPC/long-term predictive or sparse predictors) are not included; given this, it is difficult to judge how novel the contribution truly is or whether it meaningfully advances the area’s state of the art in a way that merits ICLR impact.

**Questions:**

How does the full latency budget decompose (acoustic delays, algorithmic time, and prediction lookahead) across room geometries and path variations, and what are the performance–latency trade-off curves when the causal bound is not met?

---

### Official Review · Reviewer_tPS9 · 2025-11-04

**Soundness:** 2
**Presentation:** 2
**Contribution:** 2
**Rating:** 4
**Confidence:** 2

**Summary:**

This paper proposes DeepASC, a Mamba-based architecture for Active Speech Cancellation. The method uses multi-band processing with masking mechanisms and a novel NOAS loss function. The main claim is being "the first work to actively cancel both noise and speech using deep learning." But their own experiments show existing deep learning methods already cancel both noise and speech, just with lower performance. So this might be a bit overclaimed.

**Strengths:**

1. Achieves performance improvements over baseline model such as ARN
2. Comprehensive ablation studies. Tables 9 and 10 systematically evaluate various settings such as Mamba vs LSTM vs Transformer, single-band vs multi-band, masking mechanism, dual-path structure, NOAS optimization.
3. Author shows their method is computational more efficient and also achieves better performance.
4. The real-word simulation is helpful to justify the effectiveness of the model.

**Weaknesses:**

One claim of the paper is questionable: "This is the first work to actively cancel both noise and speech using deep learning". However their own Table 7 shows DeepANC achieves ~8.56 dB on speech and ARN achieves ~10.31 dB. These are significant cancellation results. Previous methods can cancel speech, they just don't do it as well.

**Questions:**

N/A

---

### Note · Authors · 2026-01-20

I have read and agree with the venue's withdrawal policy on behalf of myself and my co-authors.